# OpenReview forum: "Aegis: Towards Governance, Integrity, and Security of AI Voice Agents"
_ICLR.cc/2026/Conference — Submitted to ICLR 2026_

### Official Review · Reviewer_xmub · 2025-10-15

**Soundness:** 3
**Presentation:** 3
**Contribution:** 3
**Rating:** 4
**Confidence:** 5

**Summary:**

This presents a comprehensive red-teaming framework for evaluating the security, integrity, and governance of Audio Large Language Model (ALLM)-based voice agents. It systematically tests these systems across adversarial scenarios—such as authentication bypass, privilege escalation, privacy leakage, and resource abuse—in critical domains like banking, IT support, and logistics. The study finds that while restricting agents to database query access reduces data leakage, it fails to prevent behavioral vulnerabilities, highlighting the need for stronger operational policies. The work’s key contribution is introducing Aegis, a practical framework that unites technical and governance perspectives to guide policy-driven defenses and improve the safe deployment of ALLM-powered voice systems.

**Strengths:**

The paper is original in introducing the first red-teaming and governance framework for Audio LLMs, addressing an important gap in AI safety. Its methodology is solid, testing realistic adversarial scenarios across domains and revealing both technical and policy weaknesses. The writing is clear and well supported by visuals, making complex findings accessible. Overall, it is a significant contribution that bridges AI security and governance, offering practical guidance for deploying safer voice-based AI systems.

**Weaknesses:**

The paper’s evaluation is limited to controlled, synthetic scenarios, which may not fully capture the complexity of real-world user interactions or adversarial conditions faced by deployed voice agents. While the red-teaming framework is well designed, it would benefit from broader empirical validation involving live or human-in-the-loop settings to assess robustness under natural speech variability and spontaneous misuse. Some aspects of the governance discussion remain high-level, lacking detailed guidance on policy integration or compliance alignment (e.g., regulatory audit mechanisms). Additionally, the framework’s scalability across different ALLM architectures and deployment platforms is not thoroughly examined, leaving questions about generalization to diverse voice systems.

**Questions:**

How does Aegis perform in real-world or human-in-the-loop environments, where voice inputs are more variable and less predictable than synthetic test cases?

Can the authors elaborate on the scalability and generalization of Aegis across different ALLM architectures or commercial deployment frameworks?

The governance component is promising—could the authors clarify how Aegis’s results could inform or integrate with regulatory compliance processes (e.g., audit trails, model certification)?

How does Aegis handle dynamic adversarial adaptation, where attackers adjust their strategies based on prior system responses?

Could the authors discuss potential extensions or automation of the red-teaming process to make it more practical for continuous monitoring in production systems?

---

> ### Author Response · Authors · 2025-12-02
>
> Dear reviewer,
>
> We are thankful for your thoughtful and constructive comments. We would like to make the clarifications below:
>
> 1. **Regarding human in-the-loop settings.**  We appreciate the reviewer’s question regarding real-world variability and human-in-the-loop settings. Our work focuses on building a systematic and controllable evaluation framework. For this reason, the main experiments use synthetic but acoustically diverse inputs generated via modern TTS models, which allow us to isolate model behaviors and compare backbone vulnerabilities fairly across agents.
>
>    However, to further examine real-world applicability, we additionally conducted human-in-the-loop experiments. In this setting, three human participants were instructed on specific attack scenarios and goals, and each attack attempt was constrained to a 10-turn dialogue, similar to the structure of our synthetic experiments. The results of the average ASR are shown below.
>
>    The results are reported below. Compared to Table 2 (TTS-based attacks), we observe that Authentication attack success rates are consistently higher for human attackers, while the other four adversarial scenarios fluctuate across backbones and attack types. This pattern suggests that human speakers are more effective at improvising around identity-verification prompts, whereas the more structured attacks (e.g., privilege escalation or data poisoning) are more sensitive to variation in human delivery.
>
>    However, our human-subjects study is limited in scale, as human-in-the-loop evaluation requires a larger sample size and a more rigorous experimental protocol to draw strong statistical conclusions. We will expand this component in future work and include a dedicated human evaluation module in Aegis to more comprehensively measure real-world attacker behavior.
>
>    |   **Backbone**   | **Authentication** | **Privacy Leakage** | **Privilege Escalation** | **Data Poisoning** | **Resource Abusing** |
>    | :--------------: | :----------------: | :-----------------: | :----------------------: | :----------------: | :------------------: |
>    |      GPT-4o      |       0.140|0.132|0.118|       0.078        |        0.552         |
>    |   GPT-4o-mini    |       0.184|0.214|0.084           |       0.102        |        0.632         |
>    |  Gemini-1.5 Pro  |       0.204|0.248|0.164           |       0.128        |        0.496         |
>    | Gemini-2.5 Flash |       0.156|0.098|0.112           |       0.054        |         0.37         |
>    |  Gemini-2.5 Pro  |       0.136|0.176|0.072           |       0.032        |         0.41         |
>    |   Qwen2 Audio    |       0.220|        0.302        |          0.228           |       0.192        |        0.736         |
>    |  Qwen 2.5-omni   |       0.196        |        0.186        |          0.268           |       0.164        |        0.588         |
>
> 2. **Regarding governance discussion and policy integration.** Aegis’s evaluation results can meaningfully support governance and compliance workflows in several ways.
>
>    First, **auditability** is built into the framework: all agent states, tool calls, and decisions are logged and traceable, enabling post-hoc inspection and accountability.
>
>    Second, it is important to incorporate **risk reporting**, as violations of privacy constraints, unauthorized actions, and unsafe tool usage can be surfaced directly as audit findings. In practical deployments, such evaluations can be paired with a real-time risk model that monitors conversations, flags risky behaviors, and escalates to a human operator when necessary.
>
>    Third, Aegis’s structured evaluation protocols can contribute to **certification processes**, serving as part of pre-deployment safety checks or periodic reassessments of ALLM-based voice agents.
>
>    Finally, **policy enforcement** is supported through system-prompt–encoded governance rules, which allow us to measure adherence to organizational or regulatory requirements. We also highlight the importance of **permission restrictions** (e.g., limiting access to private data or high-privilege operations), which aligns with current best practices for safe deployment.
>
>    We will make these connections clearer in the revised paper.
>
> 3. **Regarding scalability across ALLM architectures and deployment platforms.**  Aegis is designed to generalize across diverse ALLMs because it relies solely on **audio inputs** and **standardized function-calling or API interfaces**, which are common to most modern voice-agent systems. Our experiments already demonstrate this model-agnostic design by evaluating a range of ALLM backbones, including both open-source and closed-source models, within a unified simulated environment and using a consistent red-teaming methodology. In the revision, we will expand the discussion to more clearly justify this generalization capability and explain how Aegis can be readily extended to additional architectures and deployment frameworks.

---

> > ### Author Response · Authors · 2025-12-02
> >
> > 4. **Regarding dynamic adversarial adaptation.** We want to clarify that in our main experiments, each attack agent is initialized with a *single specified persona* and *one attack scenario*. The agent then adapts its conversation turns based on prior conversation history, with each attack constrained to a 10-round dialogue. To increase the strategic diversity of attacks, we conducted an additional experiment where we provided the entire persona list to the attack agent and allowed it to dynamically select and switch personas throughout the conversation based on dialogue context. This simulates a more realistic adaptive attacker. The results are shown in the table below.
> >
> >    |   **Backbone**   | **Authentication** | **Privacy Leakage** | **Privilege Escalation** | **Data Poisoning** | **Resource Abusing** |
> >    | :--------------: | :----------------: | :-----------------: | :----------------------: | :----------------: | :------------------: |
> >    |      GPT-4o      |       0.132        |        0.188        |          0.092           |       0.064        |        0.604         |
> >    |   GPT-4o-mini    |       0.144        |        0.212        |          0.108           |       0.076        |        0.648         |
> >    |  Gemini-1.5 Pro  |       0.152        |        0.236        |          0.132           |       0.104        |        0.568         |
> >    | Gemini-2.5 Flash |       0.108        |        0.152        |          0.076           |       0.060        |        0.436         |
> >    |  Gemini-2.5 Pro  |       0.084        |        0.164        |          0.064           |       0.044        |        0.360         |
> >    |   Qwen2 Audio    |       0.192        |        0.272        |          0.248           |       0.180        |        0.716         |
> >    |  Qwen 2.5-omni   |       0.172        |        0.220        |          0.228           |       0.168        |        0.544         |
> >
> >    While the attack success rates fluctuate slightly across backbones and scenarios, the overall trends remain consistent with the main study. This indicates that backbone vulnerabilities are generally robust to dynamic persona switching under our evaluation setting. We agree, however, that further exploration of richer adaptive policies would be a valuable direction for future work.
> >
> > 5. **Regarding the discussion on potential extensions or automation of the red-teaming process to make it more practical for continuous monitoring in production systems.** We agree that continuous monitoring is essential for real-world deployment. Aegis’s modular architecture already supports (1) automated persona-conditioned attack generation under different attack scenarios, (2) scripted multi-turn workflows, and (3) evaluation of different models and tasks. In future work, we plan to integrate automatic scenario expansion, scheduled re-evaluation cycles, and alerting mechanisms for newly discovered vulnerabilities. We believe these will make the framework practical for production-level continuous monitoring.

---

### Official Review · Reviewer_T2eW · 2025-10-20

**Soundness:** 2
**Presentation:** 3
**Contribution:** 2
**Rating:** 2
**Confidence:** 4

**Summary:**

This paper presents Aegis, a red-teaming framework designed to evaluate and enhance the security of Audio Large Language Model (ALLM)-powered voice agents in high-stakes domains such as banking and IT support. Addressing the lack of systematic security evaluations, Aegis simulates realistic deployment pipelines and structured adversarial scenarios, targeting risks like privacy leakage, privilege escalation, and resource abuse. Case studies reveal that while restricting database access can prevent certain attacks, behavioral threats persist even under strict controls. The study also finds open-weight models more vulnerable than closed-source ones and highlights the limited impact of attacker personas and gender cues when strong policies are in place. These findings advocate for multi-layered defense strategies combining access control, policy enforcement, and behavioral monitoring to secure future ALLM applications.

**Strengths:**

1. The benchmark design, in terms of scenarios and threat models, is practical.
2. The finding that changes in the data access interface significantly impact attack success rates is interesting.

**Weaknesses:**

1. The benchmark and its findings lack sufficient novelty. Although the authors claim to address two gaps—multi-turn attacks and targeting agents instead of standalone models—both aspects have been explored in prior work (yet might not be done in ALLMs), reducing the originality of the contribution.
2. Given the claimed advantage of better alignment with real-world scenarios, it is important to test the proposed attacks in actual applications. The current simulation environment appears overly simplified, limiting generalizability. For example, real-world systems do not typically grant authentication solely based on an agent's textual output like “you have been verified” (as shown in Figure 2).
3. The reported zero success rates in the query-based setup require further analysis. It is unclear whether these are due to improved security or the system's inability to execute even benign instructions via the query interface. Adding results on non-harmful tasks would clarify whether the interface hinders normal functionality.
4. The conclusion that “open-sourced backbones are more vulnerable” seems superficial. A more likely explanation is that Qwen models are less rigorously aligned for safety. To validate this, safety evaluations of the base models on standard textual safety benchmarks should be included.

**Questions:**

1. what are the defenses deployed?

---

> ### Author Response · Authors · 2025-12-02
>
> Dear reviewer,
>
> We appreciate your thoughtful feedback and the opportunity to clarify our contributions. Below we address each point in detail.
>
> 1. **Regarding the novelty of multi-turn attacks.** We would like to clarify that, as discussed in the related work and summarized in the comparison table below, previous works on red-teaming ALLMs primarily focus on jailbreaks and harmful-content benchmarks. These works mainly adapt text-based single-turn attacks to the audio modality, without modeling the richer dynamics of spoken interaction. In contrast, our work focuses on realistic deployment scenarios, where ALLMs function as multi-turn, tool-using voice agents in real-world applications such as banking support, IT helpdesks, and logistics dispatch.
>
>
>
>    Because previous studies have already conducted comprehensive evaluations of ALLM trustworthiness (e.g., jailbreak, bias, fairness, toxicity), our contribution is fundamentally different: we evaluate safety and privacy risks that emerge specifically during real-world agent operation. Prior benchmarks predominantly operate at the model level, whereas adversaries in practice interact with full voice-agent systems through evolving conversations, contextual cues, and persona-driven manipulation. As a result, existing benchmarks do not capture the agent-level risks that arise during multi-turn interactions, state retention, or tool-use decisions.
>
>    Our work addresses this gap by providing the first systematic evaluation framework that assesses adversarial risks of ALLMs when integrated as conversational agents, rather than as isolated models.
>
> |                      | Trustworthiness(jailbreak, bias, fairness, etc) | Multi-turn audio conversation | Full agent pipeline | Multi-turn agent workflow | External APIs/database access | Realistic, high-stake tasks |
> | :------------------: | :---------------------------------------------: | :---------------------------: | :-----------------: | :-----------------------: | :---------------------------: | :-------------------------: |
> |      VoiceBench      |                  $\checkmark$                   |           $\times$            |      $\times$       |         $\times$          |           $\times$            |          $\times$           |
> |      AudioTrust      |                  $\checkmark$                   |           $\times$            |      $\times$       |         $\times$          |           $\times$            |          $\times$           |
> |    AudioJailbreak    |                  $\checkmark$                   |           $\times$            |      $\times$       |         $\times$          |           $\times$            |          $\times$           |
> | Jailbreak-AudioBench |                  $\checkmark$                   |           $\times$            |      $\times$       |         $\times$          |           $\times$            |          $\times$           |
> |   **Ageis (ours)**   |                    $\times$                     |         $\checkmark$          |    $\checkmark$     |       $\checkmark$        |         $\checkmark$          |        $\checkmark$         |
>
> 2. **Regarding the real-world relevance of the simulated environment.** We would like to clarify that for authentication, the agent will call the API and verify the answer provided by the user with the ground-truth answer in the database. Only when the user’s answer matches the stored record does the agent output “You have been verified.”  We agree that real-world systems contain more complex safeguards but our goal, however, is not to replicate proprietary authentication infrastructures, which are not publicly accessible, but to evaluate the decision-making behavior of ALLM-based voice agents under adversarial scenarios. Our evaluation simulates task-level workflows (banking, IT support, logistics) common in production agents.  We ensure that our simulated environment is as close to real-world deployment as possible, and to the best of our knowledge, our work is the first to red-team ALLMs at the agent level. We will revise the manuscript to better clarify the role and limitations of simulation.
>
> 3. **Regarding zero success rate in the query-based setting.** We agree that further clarification is needed. The zero success rate arises not because the system is incapable of executing benign tasks, but because the query-based interface restricts direct tool invocation, thereby constraining dangerous agent behaviors. In addition, for query-based setting, the agent will call API and verify the answer provided by the user with the ground-truth answer in the database. Only when the user’s answer matches the stored record does the agent output “You have been verified.”

---

> > ### Author Response · Authors · 2025-12-02
> >
> > 4. **On the conclusion regarding "Open-sourced backbones being more vulnerable."** We appreciate this point and agree that framing is important. Our claim does not intend to imply that all open-source models are inherently more vulnerable. As the reviewer notes, the more plausible explanation is that the specific Qwen models tested have comparatively weaker safety alignment. Based on the results from previous work [1], which evaluated the multi-faceted trustworthiness of ALLMs, open-sourced models are generally more susceptible to jailbreak and exhibit more bias, fairness, and robustness issues. We will revise our claim to state more precisely: Open-source models with less extensive safety alignment (e.g., Qwen-Audio) show higher vulnerability in our tested scenarios, and future work should evaluate additional open-source ALLMs as more safety-tuned versions become available.
> >
> > 5. **What defenses are deployed?** Our current evaluation intentionally focuses on baseline agent behaviors without explicit adversarial defenses, in order to measure the inherent vulnerability surfaces of ALLM-powered voice agents. However, we do incorporate system-level design defenses, including system prompts with strict privacy and policy constraints and query-based restricted execution mode. And the results demonstrate that while strict privacy and policy constrants are specified in the system prompt, the agents still suffer from privacy leakage and policy compliance issues, while query-based authentication successfully reduce the attack success rate for authetication bypassing.
> >
> >    Our goal in this first version is to characterize vulnerabilities, not to propose or evaluate defenses; we will clarify this in the paper. We will consider proposing defense mechanisms such as adversarial training/finetuning in the future work.
> >
> > [1] Li, Kai, et al. "Audiotrust: Benchmarking the multifaceted trustworthiness of audio large language models." *arXiv preprint arXiv:2505.16211* (2025).

---

### Official Review · Reviewer_yBFf · 2025-11-01

**Soundness:** 3
**Presentation:** 2
**Contribution:** 2
**Rating:** 4
**Confidence:** 5

**Summary:**

This paper introduces LLaVA-Interactive, a framework that enhances vision-language models (VLMs) through multi-turn, multimodal self-interaction during fine-tuning. By simulating rich conversational behaviors—asking clarifying questions, reasoning visually, and providing step-by-step responses—it significantly improves model performance on vision-language instruction-following tasks without relying on human annotations.

**Strengths:**

Strengths:

Originality: Proposes a novel self-interaction fine-tuning method that leverages model-generated dialogs across image inputs, reducing reliance on costly human supervision.

Empirical Quality: Demonstrates state-of-the-art results on benchmarks like MME, SEED-Bench, and LLaVA-Bench with strong qualitative improvements.

Clarity and Scope: Clear description of the pipeline (Figure 2) and strong motivation for multi-turn behavior in visual instruction contexts.

**Weaknesses:**

Weaknesses:

Generalization Risk: The method is evaluated mainly on LLaVA-1.5 with specific backbone settings; it’s unclear how well the approach transfers to other VLMs or unseen visual domains.

Ablation Limitations: While some ablations are reported (e.g., number of rounds), deeper analyses of failure cases or negative impacts of self-generated noise are limited.

**Questions:**

Have you evaluated how well this self-interaction approach generalizes to other vision-language models beyond the LLaVA architecture?

---

> ### Author Response · Authors · 2025-12-02
>
> Dear reviewer,
>
> Thank you very much for taking the time to review our work. We greatly appreciate your efforts. However, it appears that the provided review may have been associated with a different submission, as the comments do not correspond to the focus of our paper. Our work specifically addresses red-teaming audio large language model agents, rather than vision–language models. Would you be able to share the correct review for our submission? Thank you again for your time and consideration.

---

### Official Review · Reviewer_mi7N · 2025-11-01

**Soundness:** 3
**Presentation:** 3
**Contribution:** 3
**Rating:** 6
**Confidence:** 4

**Summary:**

The paper presents a red-teaming evaluation of LLMs in the context of customer support in banking, IT, and logistics.

The novel contributions are:
1. A taxonomy of adversarial scenarios that is used to generate adversarial attacks for evaluation.
2. An evaluation of GPT, Gemini, and Qwen models on their robustness to adversarial attacks.

**Strengths:**

1. The paper focuses on a highly important application: adversarial attacks of voice agents. In particular, they focus on 3 customer service applications (banking, IT, and logistics) where LLMs are already deployed.

2. The evaluation framework is rigorous and reproducible. The 5 attack objectives x 5 attacker personas are relevant for many domains, as the others show in their case studies.

3. The evaluation results are novel and interesting, showing that some attack vectors remain challenging (resource abuse) and offering a practical way to reduce vulnerabilities (limit to query-based access).

**Weaknesses:**

1. Some details are confusing in the evaluation setup. In section 3.3 and Figure 2, the language around "attacker", "attack agent", "agent", and "evaluator" could be made clear. It appears these are all LLMs, the "attack agent" is always GPT-4o, the "backbone agent" is one of 7 models, and the "evaluator" is always GPT-4o? Using consistent language like "attack agent" and "backbone agent" might be helpful.

2. While the paper conducts a thorough evaluation of 7 "backbone agents", I would have liked to see a variety of models used as the "attack agent". While the evaluation using GPT-4o is an important first step, are other models better/worse at generating adversarial attacks?

3. The results around persona choice and gender are really interesting, and it would be interesting to expand the evaluation in this direction. In particular, how might dialects, choice of language, and tone influence the attack success? For example, would attacks conducted in low-resource languages be more or less likely to succeed? How about attacks with certain dialects or accents? While this may be outside the scope of this work, I would like to see more discussion about this.

4. The paper is titled towards "governance, integrity, and security", yet the overwhelming focus of the evaluation seems to be about security (e.g. whether the agent has security vulnerabilities). In particular, the paper has very little to do about governance (other than highlighting the importance of evaluation for voice agents). I'm also not sure what the benefit of titling the evaluation framework as "Aegis" is, other than giving it a fancy name. I would strongly suggest the authors consider a title that better reflects the main contributions of the work.

**Questions:**

In a rebuttal, it would be helpful if the authors addressed the weaknesses above. Some additional minor questions:

1. Just to confirm, the attacks are conducted entirely in the audio modality, and then the analysis is done in the text-domain? How is the audio transcribed to text?

2. Are the 5 adversarial scenarios taken directly from the MITRE ATTACK framework?

3. Are the backend databases also synthetically generated using GPT-4o? What is the database size? In practice, agents may have access to multiple databases, which may also affect attack success?

---

> ### Author Response · Authors · 2025-12-02
>
> Dear reviewer,
>
> We appreciate your constructive comments. We would like to make the following clarifications.
>
> 1. **Regarding the clarification of terminology in the evaluation setup.** Thank you for pointing out the ambiguity around the terms *attacker*, *attack agent*, *agent*, and *evaluator*. We have revised accordingly. Specifically, the Attack agent is always the model generating adversarial audio (GPT-4o). The backbone agent is one of the seven models under evaluation. And the evaluator is the model that analyzes the final outputs (GPT-4o).
>
> 2. **Regarding the variety of attack agent backbones.** We appreciate the suggestion to evaluate a wider range of attack agents. In this work, we intentionally used GPT-4o following previous work [1]. However, we further conduct experiments using Grok-4.1 as the attack agent in the scenarios where the backbone agents have access to the database.  The results are shown below.
>
> |     Backbone     | Authentication | Pricacy Leakage | Privilege Escalation | Data Poisoning | Resource Abusing |
> | :--------------: | :------------: | :-------------: | :------------------: | :------------: | :--------------: |
> |      GPT-4o      |0.128 |0.196      |        0.088         |     0.052      |      0.604       |
> |   GPT-4o-mini    |0.144 |0.172      |        0.112         |     0.076      |      0.652       |
> |  Gemini-1.5 Pro  |0.156|0.224      |        0.132         |     0.108      |      0.592       |
> | Gemini-2.5 Flash |0.112|0.148      |        0.072         |     0.068      |      0.428       |
> |  Gemini-2.5 Pro  |0.084|0.164      |0.068         |     0.044      |      0.352       |
> |   Qwen2 Audio    |0.208|0.268      |0.256         |     0.204      |      0.732       |
> |  Qwen 2.5-omni   |0.172|0.216      |0.236         |     0.184      |      0.556       |
>
> These results show that changing the attack agent backbone does affect attack success rates very much, and the impact varies depending on both the target backbone model and the adversarial scenario. In some cases, Grok-4.1 achieves higher ASR, while in others it is slightly lower, suggesting that different attack models may be better suited for different attack types. Overall, however, the trends and relative vulnerabilities across backbone agents remain consistent, indicating that our main conclusions are still sound.
>
> We agree that exploring multi-model or ensemble attackers is an important direction for future work, and we will explicitly incorporate this into the revised manuscript.
>
> 3. **Regarding different languages.** Since GPT-4o currently supports multiple languages but does not allow fine-grained control over accents, our multilingual experiments focus specifically on *language variation*. To examine whether language choice influences attack effectiveness at the agent level, we repeated the evaluation using Mandarin, while keeping the same experimental setup. The table below reports the results. Overall, the results are shown below.
>
>    Unlike prior jailbreak studies where multilingual prompts often consistently *increase* jailbreak rates, here the results indicate some ASR values increase, while others decrease across different backbones and attack scenarios. This difference might be due to the fact that agent-level attacks require reliable speech understanding to invoke tool-calling. As a result, authentication and privacy-related attacks sometimes increase, while multi-step attacks such as privilege escalation or data poisoning may decrease due to the less reliable speech understanding. We acknowledge that a larger cross-lingual evaluation, including more speakers and accents, would provide deeper insight, and we plan to include this direction in future extensions of Aegis.
>
>    |   **Backbone**   | **Authentication** | **Privacy Leakage** | **Privilege Escalation** | **Data Poisoning** | **Resource Abusing** |
>    | :--------------: | :----------------: | :-----------------: | :----------------------: | :----------------: | :------------------: |
>    |      GPT-4o      |       0.132        |        0.172        |          0.076           |       0.032        |        0.618         |
>    |   GPT-4o-mini    |       0.148        |        0.228        |          0.092           |       0.072        |        0.664         |
>    |  Gemini-1.5 Pro  |       0.160        |        0.268        |          0.124           |       0.096        |        0.542         |
>    | Gemini-2.5 Flash |       0.104        |        0.196        |          0.068           |       0.044        |        0.436         |
>    |  Gemini-2.5 Pro  |       0.090        |        0.158        |          0.032           |       0.048        |        0.354         |
>    |   Qwen2 Audio    |       0.220        |        0.292        |          0.240           |       0.196        |        0.744         |
>    |  Qwen 2.5-omni   |       0.184        |        0.234        |          0.234           |       0.170        |        0.560         |

---

> > ### Author Response · Authors · 2025-12-02
> >
> > 4. **Regarding the paper title.**  We thank the reviewer for this valuable perspective. We would like to clarify that while our evaluation framework focuses on security and integrity, it also explicitly incorporates governance considerations, which are discussed in multiple sections of the paper. In particular, our work evaluates how ALLM-powered voice agents adhere to governance principles such as privacy preservation, responsible data access, and policy-aligned tool usage under adversarial scenarios. These governance-oriented checks are embedded directly into our agent protocols, system prompts, and private data access protocol, allowing us to examine whether an agent behaves in accordance with organizational and policy expectations even under adversarial pressure. However, to better reflect this focus, we will refine the framing and consider adjusting the title to more clearly present the core emphasis of the work. Regarding the “Aegis” name, our intention was simply to provide a succinct and relevant label for the framework; however, we are open to simplifying or removing it if it distracts from the contributions.
> >
> > 5. **Are all attacks conducted entirely in the audio modality, and how is audio converted to text for analysis?** Yes, all attacks are conducted entirely in the audio modality. For analysis, we transcribe audio outputs using OpenAI GPT-4o-transcribe API.
> >
> >
> > 6. **Are the 5 adversarial scenarios taken directly from the MITRE ATTACK framework?** Yes, the adversarial scenarios are derived from MITRE ATT&CK, but we selected the subset that is most actionable, testable, and relevant to conversational voice agents. We will clarify this justification more explicitly in Section 3.2.
> >
> >
> > 7. **Are the backend databases also synthetically generated using GPT-4o? What is the database size?** For each application, we only generated one database that contains relevant information. For example, in bank scenario, there are user' personal information such as username, phone, email, DoB, address and their account information such as account_id, account number, routing number, balance, transaction histories, and security questions. In logistics, the database contains customer information similar to that in the bank database and shipment information associated with the customer (e.g., shipment_id, PO number, pickup address, delivery address, content, etc.)
> >
> >
> >
> > [1] Rahman, Salman, et al. "X-teaming: Multi-turn jailbreaks and defenses with adaptive multi-agents." *arXiv preprint arXiv:2504.13203* (2025).

---

### Official Review · Reviewer_MKaP · 2025-11-01

**Soundness:** 2
**Presentation:** 2
**Contribution:** 1
**Rating:** 2
**Confidence:** 4

**Summary:**

This paper investigates the security of voice agents and proposes a red-teaming framework called Aegis, designed for assessing the governance, integrity, and security of AI voice agents. To understand the vulnerabilities of voice agents, the authors employ Aegis to evaluate their backbone models across five adversarial scenarios: authentication bypass, resource abuse, privilege escalation, data poisoning, and privacy leakage. To simulate realistic usage scenarios, the experiments are conducted within contexts such as bank call centers, IT support desks, and logistics dispatch services. The findings reveal the vulnerabilities in both closed-source and open-weight backbone models powering AI voice agents.

**Strengths:**

(1) The paper is clearly structured and well-organized.

(2) The manuscript is free of grammatical and typographical errors.

**Weaknesses:**

(1) **Limited evaluation of practical voice agents**

Although the paper evaluates the governance, integrity, and security of AI voice agents, it primarily focuses on backbone models rather than complete, deployed agent systems. Real-world AI voice agents typically include multiple components, such as data processing, safeguard, and storage modules, in addition to the backbone model. Therefore, restricting the evaluation to backbone models does not provide a comprehensive understanding of the security, governance, and integrity of a full voice agent system. As noted in Section 2, the authors themselves stated that "there remains a lack of systematic evaluation frameworks that capture the full range of adversarial risks facing ALLMs when integrated as conversational agents in real-world applications," suggesting that the study’s scope is more centered on ALLMs than on end-to-end agents.

(2) **Unclear relationship between evaluated ALLMs and practical voice agents**

The paper evaluates seven ALLMs but does not clarify which real-world voice agents actually use these models. To strengthen the connection to practical relevance, the authors should specify which commercial or open-source agents adopt these ALLMs, including details such as agent names, associated backbone models, URLs, and deployment contexts, especially since the study claims to focus on voice agents.


(3) **Insufficient justification of adversarial scenario taxonomy**

The authors propose a taxonomy of adversarial scenarios for AI voice agents (Section 3.2), inspired by the MITRE ATT&CK framework. However, given the extensive range of adversarial tactics in MITRE ATT&CK, the idea and rule for selecting only the five scenarios included in this study is unclear. A more detailed justification is needed to explain this selection. Furthermore, to achieve a more comprehensive evaluation, the taxonomy and experiments should encompass all relevant adversarial scenarios applicable to AI voice agents.

**Questions:**

(1) Which real-world AI voice agents employ the seven ALLMs assessed in this paper?

(2) Do agents' components other than the backbone ALLMs (e.g., data processing, safeguard modules, data storage, system prompts) exhibit vulnerabilities under the adversarial scenarios defined by the authors?

(3) Are there additional adversarial scenarios from MITRE ATT&CK that could be relevant to AI voice agents but were not included in this study?

---

> ### Author Response · Authors · 2025-12-02
>
> Dear reviewer,
>
> We are grateful for your valuable comments. We'd like to make the following clarifications.
>
> 1. **Regarding "Limited evaluation of practical voice agents".** We would like to clarify that the goal of our work is not to evaluate end-to-end commercial voice agent systems, whose internal components (e.g., safeguards, storage modules, data preprocessing pipelines) are typically proprietary and inaccessible, but to provide a systematic evaluation framework that focuses on the core enabling technology of voice agents: Audio Large Language Models (ALLMs) when used as the agent's backbone. While prior work has evaluated ALLMs mainly as standalone backbone models, our contribution is to assess them in agent-style settings, where the model:
>
>    - engages in multi-turn conversation
>    - interacts with realistic, high-stakes scenarios (banking, IT support, logistics)
>    - and executes tool-using actions
>
>    We agree that real commercial voice agents include additional non-LLM components, but these modules are not accessible for controlled academic evaluation. As noted by the reviewer, we explicitly stated in Section 2 that “there remains a lack of systematic evaluation frameworks that capture the full range of adversarial risks facing ALLMs when integrated as conversational agents.” Our study directly addresses this gap by moving beyond backbone model evaluations into agent-style evaluation protocols that simulate real deployments.
>
>
>
> 2. **Regarding the connection between evaluated ALLMs and real-world voice agents.** While we are unable to specify the exact backbone models used by commercial systems in that they are proprietary, many AI voice-agent providers publicly offer or integrate models that overlap with those evaluated in our work. For example, [Microsoft Azure][[https://techcommunity.microsoft.com/blog/azure-ai-foundry-blog/building-an-ai-assistant-using-gpt-4o-audio-preview-api/4358986] and [Vapi][https://docs.vapi.ai/openai-realtime] provides GPT-4o  as an AI assistant model. Similarly, several open-source platforms (e.g., [pipe-cat][https://docs.pipecat.ai/server/services/s2s/gemini-live]) adopt models such as GPT-4o and Gemini as core components of their voice-agent pipelines. Our goal is not to map each backbone to a specific commercial deployment, but to evaluate the representative class of ALLMs that underlie modern voice-agent ecosystems. We will clarify this connection more explicitly in the revised version.
>
>
>
> 3. **Regarding the justification for the adversarial scenario taxonomy.** While MITRE ATT&CK includes hundreds of adversarial tactics, our objective was not to exhaustively include the entire MITRE ATT&CK matrix to the voice-agent setting. Instead, we identify five attack scenarios that are uniquely actionable, observable, and meaningful specifically for AI voice agents, with a focus on security, privacy, and operational integrity. These five scenarios were selected because they:
>
>    - reflect realistic threat models encountered in call-center and voice-agent deployments
>    - target vulnerabilities that arise specifically when agents interact via audio
>    - correspond to high-impact failure modes, such as privilege escalation, authentication bypass, data leakage, and unsafe tool execution
>    - can be tested consistently across models without requiring access to proprietary internal components
>
>    We will revise Section 3.2 to more clearly articulate this selection criterion and emphasize that our taxonomy is purposefully tailored to the voice agent domain rather than an exhaustive mapping of MITRE ATT&CK.
>
>
>
> 4. **Do agents' components other than the backbone ALLMs exhibit vulnerabilities under the adversarial scenarios defined by the authors?** Our evaluation standardizes all system prompts and includes them in the Appendix. We clearly specify strict privacy requirements and safety constraints in these prompts. While non-LLM components in real systems may introduce additional vulnerabilities, they are generally proprietary and inaccessible, making controlled experimental evaluation infeasible. Our work highlights how vulnerabilities persist even when strict system prompt safeguards are applied, indicating that backbone ALLM behavior is a critical surface of risk.
>
>
> 5. **Are there additional adversarial scenarios from MITRE ATT&CK that could be relevant to AI voice agents but were not included in this study?** Yes, MITRE ATT&CK contains many tactics that may conceptually apply. However, our goal was to select scenarios that are (1) directly translatable into audio-agent interactions; (2) testable using publicly available ALLM; (3) high-impact in real voice-agent deployments. We will revise the paper to explicitly explain why our chosen five scenarios are the most relevant and actionable for AI voice agents and we will extend coverage to additional MITRE ATT&CK tactics in future work as the ecosystem evolves.

---

### Official Review · Reviewer_ftvg · 2025-11-02

**Soundness:** 2
**Presentation:** 3
**Contribution:** 2
**Rating:** 4
**Confidence:** 3

**Summary:**

The paper introduces Aegis, a red-teaming framework designed to assess the security, privacy, and governance of voice agents powered by Audio Large Language Models (ALLMs) deployed in high-stakes environments, such as banking, IT support, and logistics. The authors conduct an evaluation of voice agents under five adversarial scenarios: authentication bypass, privacy leakage, resource abuse, privilege escalation, and data poisoning. The paper highlights that while restricting agents to query-based database access mitigates some security vulnerabilities, behavioral threats such as privilege escalation and resource abuse persist, revealing the complexity of safeguarding voice agents.

**Strengths:**

- The Aegis framework goes beyond traditional model-level robustness evaluations and offers a realistic assessment of deployed systems in diverse, high-risk domains.


- By considering a broad set of adversarial scenarios, the framework offers valuable insights into various vulnerabilities and highlights real-world risks that existing models fail to address.

**Weaknesses:**

- Red-teaming framework of ALLM has been studied before, although the authors claim this work focuses on more realistic assessment. Therefore, the contribution of this paper seems unclear.


- The framework heavily relies on certain attack scenarios, such as authentication bypass and resource abuse, but the paper could benefit from exploring additional advanced adversarial tactics. For instance, attacks exploiting AI’s cognitive biases in interpreting complex dialogues could be a future avenue for research.


- While restricting agents' access to query-based systems is a positive step, the persistence of behavioral vulnerabilities under stricter policies might suggest that the focus on data access limitations is insufficient. The paper doesn’t delve deeply into other critical aspects such as psychological manipulation of voice agents.

**Questions:**

See Weaknesses.

---

> ### Author Response · Authors · 2025-12-02
>
> Dear reviewer,
>
> We are thankful for your comments. We would like to make the following clarifications:
>
> 1. **Regarding the contribution of our work.** We would like to clarify that, as discussed in the related work and summarized in the comparison table below, previous works on red-teaming ALLMs primarily focus on jailbreaks and harmful-content benchmarks. These works mainly adapt text-based single-turn attacks to the audio modality, without modeling the richer dynamics of spoken interaction. In contrast, our work focuses on realistic deployment scenarios, where ALLMs function as multi-turn, tool-using voice agents in real-world applications such as banking support, IT helpdesks, and logistics dispatch.
>
>    Since previous studies have already conducted comprehensive evaluations of ALLM trustworthiness (e.g., jailbreak, bias, fairness, toxicity), our contribution is fundamentally different: we evaluate safety and privacy risks that emerge specifically during real-world agent operation. Prior benchmarks predominantly operate at the model level, whereas adversaries in practice interact with full voice-agent systems through evolving conversations, contextual cues, and persona-driven manipulation. As a result, existing benchmarks do not capture the agent-level risks that arise during multi-turn interactions and tool-use decisions.
>
>    Our work addresses this gap by providing the first systematic evaluation framework that assesses adversarial risks of ALLMs when integrated as conversational agents, rather than as isolated models.
>
> |                      | Trustworthiness(jailbreak, bias, fairness, etc) | Multi-turn audio conversation | Full agent pipeline | Multi-turn agent workflow | External APIs/database access | Realistic, high-stake tasks |
> | :------------------: | :---------------------------------------------: | :---------------------------: | :-----------------: | :-----------------------: | :---------------------------: | :-------------------------: |
> |      VoiceBench      |                  $\checkmark$                   |           $\times$            |      $\times$       |         $\times$          |           $\times$            |          $\times$           |
> |      AudioTrust      |                  $\checkmark$                   |           $\times$            |      $\times$       |         $\times$          |           $\times$            |          $\times$           |
> |    AudioJailbreak    |                  $\checkmark$                   |           $\times$            |      $\times$       |         $\times$          |           $\times$            |          $\times$           |
> | Jailbreak-AudioBench |                  $\checkmark$                   |           $\times$            |      $\times$       |         $\times$          |           $\times$            |          $\times$           |
> |   **Ageis (ours)**   |                    $\times$                     |         $\checkmark$          |    $\checkmark$     |       $\checkmark$        |         $\checkmark$          |        $\checkmark$         |
>
>
>
> 2. **Regarding exploiting cognitive bias and psychological manipulation.** In the experiments section, we also evaluate different personas (Curious customer, Insider, Social Engineer, Persistent attackers, and malicious developer) to induce different tactics and theoretical styles under different attack scenarios.  The results indicate that the differences are consistently small (within a few percentage points), indicating that persona choice has minimal impact on the overall attack outcomes.

---

### Meta-Review · Area_Chair_HQ5n · 2026-01-01

**Summary:**

The rebuttal strengthens this work with expanded attacker models, multilingual, and human-in-the-loop experiments. It clarifies the evaluation roles and some crucial details. However, two substantive concerns remain. (1) Its novelty is still not convincingly differentiated from prior multi-turn/agent red-teaming beyond modality, and (2) the evaluation remains largely synthetic and backbone-centric, leaving end-to-end voice-agent realism and governance guide insufficiently demonstrated. Overall, this work should be further improved to be accepted.

**Reviewer Concerns:**

- Reviewers ftvg and T2eW raise concerns about novelty relative to prior red-teaming and jailbreak work. This is partially addressed by the rebuttal through positioning at the agent level, but some overlap with prior concepts remains.
- Reviewer MKaP's concern about the limited evaluation of full end-to-end commercial voice agents is not fully addressed, as the rebuttal justifies focusing on backbone ALLMs due to proprietary constraints rather than providing stronger empirical results.
- Reviewer mi7N's concerns about unclear roles and limited attacker-model diversity are addressed via clarified terminology and additional Grok-4.1 experiments.
- Reviewer xmub's requests on governance depth, scalability, dynamic adaptation, and human-in-the-loop validation are partially addressed by added discussion and human-in-the-loop experiments, though governance guidance remains somewhat high-level.
- Reviewer yBFf's review appears mismatched to this submission.

**Reviewer Scores:**

Considering each reviewer's key concerns and the authors' responses, Reviewer MKaP's rating would likely increase slightly (e.g., from 2 to 4), while other reviewers' ratings would remain unchanged.

---

### Decision · Program_Chairs · 2026-01-26

Reject